# Measuring financial risk protection in health benefits packages: scoping review protocol to inform allocative efficiency studies

Gerard Joseph Abou Jaoude, Jolene Skordis-Worrall, Hassan Haghparast-Bidgoli

Institute for Global Health, University College London, London, UK

**Correspondence to**
Mr Gerard Joseph Abou Jaoude; gerard.jaoude.15@ucl.ac.uk

## ABSTRACT

**Introduction** To progress towards Universal Health Coverage (UHC), countries will need to define a health benefits package of services free at the point of use. Financial risk protection is a core component of UHC and should therefore be considered a key dimension of health benefits packages. Allocative efficiency modelling tools can support national analytical capacity to inform an evidence-based selection of services, but none are currently able to estimate financial risk protection. A review of existing methods used to measure financial risk protection can facilitate their inclusion in modelling tools so that the latter can become more relevant to national decision making in light of UHC.

**Methods and analysis** This protocol proposes to conduct a scoping review of existing methods used to measure financial risk protection and assess their potential to inform the selection of services in a health benefits package. The proposed review will follow the methodological framework developed by Arksey and O'Malley and the subsequent recommendations made by Levac *et al*. Several databases will be systematically searched including: (1) PubMed; (2) Scopus; (3) Web of Science and (4) Google Scholar. Grey literature will also be scanned, and the bibliography of all selected studies will be hand searched. Following the selection of studies according to defined inclusion and exclusion criteria, key characteristics will be collected from the studies using a data extraction tool. Key characteristics will include the type of method used, geographical region of focus and application to specific services or packages. The extracted data will then be charted, collated, reported and summarised using descriptive statistics, a thematic analysis and graphical presentations.

**Ethics and dissemination** The scoping review proposed in this protocol does not require ethical approval. The final results will be disseminated via publication in a peer-reviewed journal, conference presentations and shared with key stakeholders.

### Strengths and limitations of this study

► This protocol proposes to conduct the first scoping review of methods used to measure financial risk protection.
► The proposed review aims to outline financial risk protection measurement methods and detail their application to specific services or packages to assess how they could be included in allocative efficiency modelling.
► The review will be carried out by the corresponding author only, under the supervision of coauthors, which introduces a greater margin for error and potential bias for the screening and data extraction stages.
► While the strengths, weaknesses and applications of different methodologies will be discussed, the quality of included studies will not be considered.
► A consultation stage is proposed in this protocol, which can improve the comprehensiveness and relevance of the final review.

globally, as highlighted by the inclusion of UHC as target 3.8 in the UN Sustainable Development Goals (SDGs).[1] According to WHO, 'UHC means that all individuals and communities receive the health services they need without suffering financial hardship'.[2] This definition by WHO is not prescriptive but is clear in outlining that UHC does not involve providing all possible health services free of charge for everyone. Instead, UHC entails the provision of a defined package of effective services that are of sufficient quality and maximise health, equity and protection from financial hardship.[3–5]

Given the resource constraints that countries face, particularly low and middle-income countries, challenging decisions will need to be made on what services will be provided free of charge and which services will not.[6 7] When working towards UHC, governments will therefore have to define a

## INTRODUCTION
### Background
The political and academic focus on Universal Health Coverage (UHC) continues to grow

health benefits package that is free at the point of use, sometimes referred to as a basic or essential package of health services. Health financing reforms, and how best to implement selected services to maximise quality and access, will also need to be considered.[6–10] Best practice in priority setting commonly involves a transparent, systematic and evidence-based approach with high levels of stakeholder involvement.[6 11] Due to financial and human resource constraints, countries seeking to embark on these complex health system reforms will require administrative and analytic support for evidence-based decision making.[6 12 13]

Modelling tools can help support national analytical capacity and facilitate evidence-based priority setting.[6 14 15] Allocative efficiency analyses may be especially useful in this context as they are able to estimate and compare the health impact of various packages of services, with different budgets, while considering trade-offs between desired objectives—for example, minimising disease incidence or mortality.[16 17]

To date, allocative efficiency analyses have commonly been used to inform decision making for national disease control programmes, such as national tuberculosis or HIV programmes.[15 18 19] There is largely unexplored potential for allocative efficiency analyses to inform priority setting across diseases, while considering the costs, effectiveness, equity impact and financial risk protection of health benefits packages. The literature on how to consider costs and effectiveness is well established, but there is less research on the inclusion of equity measures,[20–22] and to our knowledge, financial risk protection has not yet been incorporated into allocative efficiency modelling analyses.

## Rationale

Links between poverty and health have been studied extensively, highlighting the importance of socio-economic status and income inequality as determinants of health.[23–25] Likewise, indebtedness due to medical expenses is one of the main pathways into and a major cause of remaining trapped in poverty.[26–28] For individuals or households, particularly those with low income, medical expenses due to health shocks can put at risk or negatively impact non-medical consumption causing financial strain or hardship.[29 30] Financial risk protection can therefore be understood as protection against the latter by the partial or full subsidisation of healthcare costs through mechanisms such as public financing and formal, informal or self-provided insurance.[31] Several measures of financial risk protection currently exist, but the most frequently used indicators, adopted both by WHO and the World Bank, are measures of financial hardship and can be categorised into catastrophic health expenditure and impoverishment. Catastrophic health expenditure occurs when the proportion of a household's medical expenses relative to its income or capacity to pay exceeds a defined threshold.[32–34] Measures of impoverishment on the other hand, aim to capture the number of

households pushed into, or deeper into, poverty due to healthcare costs.[34 35]

It is estimated that 808 million people incurred catastrophic health expenditures in 2010.[36] Financial risk protection to avert catastrophic health expenditure and medical impoverishment is a core component of UHC. It is explicitly referred to in target 3.8 of the SDGs, but countries are not on track to meet targets for financial risk protection.[37] When defining or updating national health benefits packages, governments should therefore consider financial risk protection as a key dimension of the priority setting process alongside the maximisation of health benefits. If allocative efficiency and modelling analyses are to inform policies for UHC and the contents of health benefits packages, a method to estimate financial risk protection must be incorporated into the analyses. A scoping review of financial risk protection measurement methods and how they could be applied to allocative efficiency analyses can provide a foundation for modelling efforts to become more relevant in light of the ongoing push towards UHC. No such review has yet been conducted, and this protocol proposes to address this gap.

## Objectives

The primary objective of the proposed scoping review is to outline the different methods used to measure financial risk protection and discuss their strengths and weaknesses as well as application to specific services or health packages. Subsequently, the review will seek to consider how existing financial risk protection measurement methods could be applied to allocative efficiency analyses of:
1. Disease-specific health packages.
2. Health benefits packages that address multiple diseases.

The secondary objective of the review will be to build on the work carried out by the World Bank and WHO to track progress towards UHC and financial risk protection.[38–41] The review will compare the amount of research on financial risk protection by country and region against the burden of financial hardship to establish any mismatch between research focus and burden.

## METHODS AND ANALYSIS

Scoping reviews benefit from a breadth of content by providing an overview of all the research in a given subject area. They also achieve depth through the mapping and interpretation of research.[42] The objectives of the proposed review lend themselves well to such an approach. They are broad in their scope but precise in their aim to map existing research on financial risk protection measurement methods while considering how they could be applied to allocative efficiency analyses. The proposed scoping review will follow the six stages outlined in the framework by Arksey and O'Malley,[42] hereinafter referred to as Arksey and O'Malley's framework, which are discussed in detail below. The recommendations

by Levac *et al*,[43] who further clarify and build on Arksey and O'Malley's framework, will also be incorporated throughout the review process. The reporting of the review will comply with the Preferred Reporting Items for Systematic Reviews and Meta-Analyses checklist recently developed for scoping reviews.[44]

## Stage 1: Identifying the research questions

The overarching question of the proposed review will guide the search strategy and the interpretation and reporting of the results: what are the different methods currently used to measure financial risk protection? This question should be broad enough to ensure that all the existing literature is captured and analysed in the review. Two subquestions have also been identified, which will focus on the application of existing methods to health services or packages and their potential to be included in allocative efficiency analyses. The two subquestions are as follows: (1) how have different methods been used to estimate financial risk protection for health services or packages? and (2) how could existing financial risk protection measurement methods be applied to allocative efficiency analyses? These questions will be subjected to iterative thinking as the review is carried out. While the overarching question is unlikely to change, the subquestions proposed may be refined or additional questions included.

## Stage 2: Identifying relevant studies

Stage 2 of the scoping review will involve identifying relevant studies for selection. A systematic search strategy, which is guided by the overarching question and its subcomponents, will be used to carry out this stage of the review. The following electronic databases will be searched: (1) PubMed; (2) Scopus; (3) Web of Science and (4) Google Scholar. The initial set of search terms and strategy proposed in table 1 and online supplementary file 1 respectively, generates 1594 results on PubMed. The search terms and search strategy have been formulated to identify research on the measurement of financial risk protection and how the different methods have been applied. Search results will be downloaded and imported using EndNote, and duplicates will be deleted prior to screening according to the inclusion and exclusion criteria set in stage 3.

Relevant grey literature on financial risk protection and UHC published by institutions or organisations working on these topics will also be scanned. Examples of such institutions or organisations are the WHO, World Bank, Disease Control Priorities Network, Centre for Global Development, International Decision Support Initiative, Health Policy Plus, UN Department of Economic and Social Affairs and the Health Intervention and Technology Assessment Program. The reference lists of all literature deemed relevant according to the inclusion and exclusion criteria from stage 3 will then be hand-searched. In line with Arksey and O'Malley's framework and Levac *et al*, the search terms and search strategy will be improved in an iterative process as we become more familiar with the literature.

| Table 1 List of search terms and search strategy | |
| --- | --- |
| **Search terms for financial risk protection** | **Search terms for methods used and their application** |
| "Financial risk protection" OR | "Universal health coverage" OR |
| "Financial hardship " OR | "UHC" OR |
| "Financial protection" OR | "Health benefits package" OR |
| "Financial protection in health" OR | "HBP" OR |
| "Catastrophic health expenditure" OR | "Basic package of health services" OR |
| "Catastrophic medical expenditure" OR | "BPHS" OR |
| "Catastrophic health expenditure risk" OR | "Essential package of health services" OR |
| "Catastrophic medical expenditure risk" OR | "EPHS" OR |
| "Catastrophic health payment" OR | "Priority setting" OR |
| "Catastrophic medical payment" OR | "Health policy" OR |
| "CHE" OR | "Resource allocation" OR |
| "CMP" OR | "Allocative efficiency" OR |
| "Medical induced poverty" OR | "Methodology" OR |
| "Health induced poverty" OR | "Measurement" OR |
| "Payment-induced poverty" OR | "Modelling" OR |
| "Catastrophic payment" OR | "Distributional analysis" OR |
| "Catastrophic cost" OR | "Tracking" OR |
| "Health Impoverishment" OR | "Monitoring" OR |
| "Medical Impoverishment" OR | "Estimating" OR |
| "Extended cost-effectiveness analysis" OR | "Quantifying" OR |
| "ECEA" | "Threshold" OR |
| | "Healthcare financing" OR |
| | "Health insurance" OR |
| | "Social health insurance" OR |
| | "National health insurance" |
| | "National health programs" |

## Stage 3: Selecting studies

The third stage of the review will involve setting inclusion and exclusion criteria. Preliminary inclusion and exclusion criteria are proposed below, but these will be subjected to iterative thinking as specified by Arksey and O'Malley. Studies generated by the search will then be screened according to the inclusion and exclusion criteria in a two-step selection process. The first screen will be based on the relevance of the titles and abstracts, followed by a second screen after reading the remaining articles in full.

### Inclusion criteria

Peer-reviewed journal articles and grey literature measuring or discussing methodologies for measuring financial risk protection will be included. Research estimating or conceptualising financial risk protection using either quantitative or qualitative techniques, regardless of the year published, geographic location, disease area of focus, health services or population groups considered will also be included.

### Exclusion criteria

Papers that lack discussion either on the methodology behind the measures, their application to health services or implications for priority setting will be excluded. Papers that cannot be accessed through institutional log-in or in languages other than English and French will also be excluded.

## Stage 4: Charting the data

The data from studies identified and selected for final inclusion in stage 3 will be collected and charted. A data collection tool will be developed to collect key characteristics from the journal papers and grey literature included, such as the methods used, application to health services or packages and country or region of interest. The tool will be developed in an iterative manner during the data collection process. Any decisions to amend the tool during the data collection process and any changes to the overall review protocol will be documented and reported appropriately. Both descriptive statistics and qualitative analyses will then be carried out on the data collected before reporting the results.

Data will be aggregated and charted according to the different characteristics collected. For example, the number of studies conducted by or concerned with a given type of method or by geographic region of interest. While there is an ongoing initiative to track progress towards UHC and financial risk protection on a global scale,[38–41] this does not map the amount of research conducted by geographic region. Therefore, using the data extracted, we will attempt to compare the amount of research conducted on financial risk protection with the burden of financial hardship by geographic region. Given that the focus of the proposed review is on mapping the different methodologies available to measure financial risk protection, a thematic analysis will also be undertaken to summarise and better understand the different approaches and any overlap or links between them. The strengths, weaknesses and the applications of different methodologies will then be discussed and investigated in the review based on the data extracted.

## Stage 5: Collating, summarising and reporting the results

The data collected in stage 4 will be collated, summarised and reported to effectively map the different methods currently used to measure financial risk protection. Papers on the different methods used to measure financial risk protection will be classified according to the different characteristics identified and gathered through data extraction and charting. Tables and graphical presentations such as bar charts will be used to present the data in an aggregated manner to provide an overview of the methods and how they could be included in allocative efficiency modelling. The latter will be assessed according to certain characteristics collected for different measurement methods, such as data requirements and application to individual health services or packages.

## Stage 6: Consultation

The consultation stage aims to involve specialists in financial risk protection and allocative efficiency modelling, as these will likely be the primary target audience of the work.

The consultation stage was described as optional but desirable by Arksey and O'Malley. Since then, the recommendations outlined by Levac *et al* have suggested this should be a required step as it adds methodological rigour. In line with the recommendation to establish a clear purpose for the consultation, the following two objectives are proposed due to their potential to strengthen the review. First, to ascertain whether any methods or approaches to measure financial risk protection have not been captured by the review. Second, to assess the accuracy and instructiveness of the review and the results reported.

### Study timeline

The first and second stages of this study started in June 2018 and were required to develop this protocol. Stage 3 of the review commenced in October 2018 and took 5 months to complete by February 2019. The hand-searching of included articles and the fourth stage are expected to take 4 months and finish by June 2019. An additional 2 months have been assigned for stages 5 and 6 and for reporting the results. The estimated completion date of this scoping review is therefore August 2019.

### Patient and public involvement

Patients and the public were not involved in the development of this protocol. If specialists agree to be involved in the consultation stage, details of this will be included in the final scoping review.

## ETHICS AND DISSEMINATION

On completion of all the steps outlined in this protocol, the findings of the proposed scoping review will be disseminated through the submission of a paper for peer-reviewed publication and conference presentations. The findings will also be shared with WHO, the Disease Control Priorities Network and other organisations working to implement and estimate the impact of health benefits packages in a move towards UHC or that are involved in allocative efficiency modelling. This study does not require ethical approval as only secondary data will be used.

The proposed scoping review will be the first to comprehensively map the status quo of research on the measurement of financial risk protection, both in terms of methodological developments and empirical work. The extraction of data from included literature is another advantage given that data extraction does not always feature in scoping reviews.[45] However, there are key limitations to the proposed review. First, the different stages have and will continue to be undertaken by the corresponding author only, under the supervision of coauthors. This introduces a greater margin for error and potential bias, particularly for stages 3 and 4.[46 47] Second, while the strengths, weaknesses and applications of different methodologies will be discussed, the quality of included studies will not be considered as would commonly be the case in systematic reviews.[45] Third, the review will fail to capture some relevant empirical work published in languages other than English or French.

The global movement towards UHC is accelerating, and as governments undertake policy reforms it is essential that the decisions made are transparent, systematic and evidence-based. Allocative efficiency analyses can inform such reforms through a systematic and evidence-based approach. However, measures of financial risk protection will need to be incorporated into allocative efficiency modelling if the efforts are to be useful and relevant to decision makers. The proposed scoping review outlined in this protocol can enable the latter by mapping the different available approaches for measuring financial risk protection and discussing their suitability for analyses seeking to inform priority setting in health.

**Contributors** All authors contributed meaningfully to the preparation, drafting and editing of this scoping review protocol. GJAJ and HH-B conceived the idea and GJAJ led protocol development. GJAJ (corresponding author) conceptualised the research questions and core research plan details before preparing the initial draft of this manuscript. JS-W and HH-B provided critical input on the draft, research questions and methodology. All authors approved the final submitted manuscript after several iterations and rounds of editing and agree to be accountable for all aspects of this protocol.

**Funding** This work was supported by the Bill and Melinda Gates Foundation grant number OPP1179683.

**Competing interests** None declared.

**Patient consent for publication** Not required.

**Provenance and peer review** Not commissioned; externally peer reviewed.

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
