## [Reviewer comments · BMJ Open]

ARTICLE DETAILS

TITLE (PROVISIONAL)	Measuring Financial Risk Protection in Health Benefits Packages: A Scoping Review Protocol to Inform Allocative Efficiency Studies
AUTHORS	Abou Jaoude, Gerard; Skordis-Worrall, Jolene; Haghparast-Bidgoli, Hassan

VERSION 1 - REVIEW

REVIEWER	Zachary Olson UC Berkeley, USA
REVIEW RETURNED	04-Jan-2019

GENERAL COMMENTS	This is a very strong protocol and I only have two comments which relate to the limitations of your study. 1. Your overall research question and first sub-question are clear and addressable with your strategy. However, I am curious as to how you will effectively narrow the articles and conclusions down for your second sub-question. The type of literature you will get from Table 1, column B will be quite diverse and not necessarily have anything to do with allocative efficiency. It might be useful to limit the search terms here, or also include financial risk protection as a term to assess how it has played a role in allocative efficiency to date. Your inclusion criteria seem to apply exclusively to column A of table 1. If you intend to apply it to Column B, then you are pulling significantly more literature than is necessary.2. Do you expect to miss any literature from developing countries because they are not in English/French?
--

REVIEWER	Kristine Husøy Onarheim Department of Global Public Health and Primary Care, University of Bergen, Norway
REVIEW RETURNED	01-Mar-2019

GENERAL COMMENTS	The manuscript is a protocol on a scoping review to inform allocative efficiency studies. The study aims to study methods used to measure FRP, how these are used and how these measurements may be used. As outlined in the protocol, there is a gap in the current literature on how FRP is measured across contexts. A better understanding of how these are used is necessary to make sure that FRP is considered as countries moves towards UHC. The study design seems fitting and is well-
--

planned. The study is likely to provide an evidence-informed platform for next steps relevant for researchers working on UHC as well as decision making and planners defining essential health packages.

However, there are some limitations that should be addressed to improve the paper:

Introduction

1) Please provide a clear definition of how you understand FRP early in the manuscript. While different versions exist, it is crucial that you explicitly define how you understand that term.

2) Your description of UHC and how it is closely linked to FRP is well written, but focus on technical terms. For the uninformed reader, please add a short sentence to the reader of why FRP matters (poverty – health interlinkages).

3) Paragraph three under background (Page 2, line 30-35) is describes how modelling tools are relevant for priority setting, with focus on health impacts. Please describe why information on FRP is needed here (□ to be considered alongside health benefits)

4) Line 40, page: The details added on ref 20 are probably relevant, but is difficult to understand unless more information is added

Objectives

The objectives are specific and understandable. The objectives link closely to the section Stage 1, but seems to differ slightly. To avoid confusion, I strongly encourage you to stick to same objectives in the abstract, objectives and Stage 1 sections.

I think your focus in Section one, which links the question of how measures have been used to the question of how measures could be used, is of particular interest to the reader. If possible, it would be interesting to know the strengths and limitations of the different approaches (could be included in the data you extract, followed by a discussion/analysis by the authors). Whether there are differences between regions or across diseases is of academic interest, but not the most important question (in my view).

Methods and analysis

Methods are explained in detail. The authors may clarify the following points:

1) Why was a scoping review chosen (not a systematic)?

2) Your choice of framework (Arksey and O'Malley) seems fitting. However, please clarify why this was chosen. Also, why have you chosen to not use the PRISMA guidelines

(<https://annals.org/aim/fullarticle/2700389/prisma-extension-scoping-reviews-prisma-scr-checklist-explanation>). I think you address most of their points, but it would interest the reader to understand the rationale for your chosen approach.

3) When you use the Arksey and O'Malley framework, please clarify how it is special (and why it is relevant for your study)

4) Stage 2: Adding information from the grey literature seems fitting. In the paper you list highly relevant institutions. To explicitly allow others to understand which grey literature you search, I would suggest that you focus on key publications from these institutions (on UHC and/or FRP).

5) Study timeline: The section needs to be revised before publication of the protocol.

	6) Is it one person who will conduct the analysis and data extraction? If so, please clarify. Discussion Well-written. You may also provide the reader with a discussion of the strengths and limitations of the study.
--	---

VERSION 1 – AUTHOR RESPONSE

Reviewer 1	
Comment	Response
This is a very strong protocol and I only have two comments which relate to the limitations of your study.	We thank the reviewer for their kind appraisal of our work.
1. Your overall research question and first sub-question are clear and addressable with your strategy. However, I am curious as to how you will effectively narrow the articles and conclusions down for your second sub-question. The type of literature you will get from Table 1, column B will be quite diverse and not necessarily have anything to do with allocative efficiency. It might be useful to limit the search terms here, or also include financial risk protection as a term to assess how it has played a role in allocative efficiency to date. Your inclusion criteria seem to apply exclusively to column A of table 1. If you intend to apply it to Column B, then you are pulling significantly more literature than is necessary.	We agree that the primary objective of the review is to assess the transferability of existing financial risk protection (FRP) measurement methods to allocative efficiency analyses. Our apologies that this was unclear in the initial submission, but the secondary objective of the review (please see changes to pg.4 lines 30-42) will be to generate a geographic map to compare research conducted on FRP with the burden of financial hardship as monitored by the WHO and World Bank. For this reason, our search terms and inclusion criteria are broad in order to capture all empirical research as well as methodological which we agree would have required more restrictive inclusion criteria. A key strength of scoping reviews is that they chart the parameters of existing research in a given field of study. Mapping not only existing methodologies but also empirical research on FRP measurement can help better guide future work, and there are precedents for the proposed secondary objective in the field of cost-utility analyses (https://journals.sagepub.com/doi/abs/10.1177/0272989X05276853). With regards to narrowing down the literature to effectively address the primary objective, existing FRP measurement methods will be mapped followed by the use of several criteria to determine the transferability of each to allocative efficiency analyses. The criteria will consider the complementarity of: (a) how populations are defined, (b) types of data required, and (c) types of metrics generated, among others, between FRP measurement methods and commonly used allocative efficiency models.
2. Do you expect to miss any literature from developing countries because they are not in English/French?	The study selection process started in November 2018 and has been finalised in the case of PubMed articles. There are just under 15 papers so far for which there are no equivalents in English. All of these have been empirical from South America (12 journal articles so far) and East Asia (2 so far). That said, while the exclusion of these papers will result in some empirical research being missed for the global research map, all research on the methods to measure financial risk protection should be captured and effectively address the review's primary objectives. (please see changes to pg.8 lines 45-46).
Reviewer 2	
Comment	Response

The manuscript is a protocol on a scoping review to inform allocative efficiency studies. The study aims to study methods used to measure FRP, how these are used and how these measurements may be used. As outlined in the protocol, there is a gap in the current literature on how FRP is measured across contexts. A better understanding of how these are used is necessary to make sure that FRP is considered as countries moves towards UHC. The study design seems fitting and is well-planned. The study is likely to provide an evidence-informed platform for next steps relevant for researchers working on UHC as well as decision making and planners defining essential health packages. However, there are some limitations that should be addressed to improve the paper:	We thank the reviewer for their kind appraisal of our work.
Introduction 1) Please provide a clear definition of how you understand FRP early in the manuscript. While different versions exist, it is crucial that you explicitly define how you understand that term. 2) Your description of UHC and how it is closely linked to FRP is well written, but focus on technical terms. For the uninformed reader, please add a short sentence to the reader of why FRP matters (poverty – health interlinkages). 3) Paragraph three under background (Page 2, line 30-35) describes how modelling tools are relevant for priority setting, with focus on health impacts. Please describe why information on FRP is needed here (-> to be considered alongside health benefits) 4) Line 40, page: The details added on ref 20 are probably relevant, but is difficult to understand unless more information is added	1-2) We hope that the paragraph added at the start of the 'Rationale' section (pg. 3 lines 42-47, and pg. 4 lines 1-12) addresses points 1 and 2 that you have raised. 3) The need for FRP to be included in allocative efficiency settings is covered in paragraph 2 of the 'Rationale' section. We feel a discussion of this is better reserved for the 'Rationale' section given that it forms the rationale not only for including FRP in allocative efficiency analyses but also for the review itself. 4) We agree, the ending of the sentence (pg. 3 line 31-32) was not needed and was a repetition of pg.3 lines 36-38 that follow. Reference 20 has therefore been merged with the literature referred to in pg. 3 lines 36-38.

Objectives The objectives are specific and understandable. The objectives link closely to the section Stage 1, but seems to differ slightly. To avoid confusion, I strongly encourage you to stick to same objectives in the abstract, objectives and Stage 1 sections. I think your focus in Section one, which links the question of how measures have been used to the question of how measures could be used, is of particular interest to the reader. If possible, it would be interesting to know the strengths and limitations of the different approaches (could be included in the data you extract, followed by a discussion/analysis by the authors). Whether there are differences between regions or across diseases is of academic interest, but not the most important question (in my view).	We agree about the importance of investigating the strengths, weaknesses and applications of different FRP measurement methods. Our apologies that this was not clear in the first draft submitted, and hopefully the added lines 31-32 on pg.4 and lines 31-33 on pg.7 have made it explicit that these will be discussed in and a main focus of the proposed review. We also agree that the main objectives of the review are to identify measurement methods that are potentially applicable to allocative efficiency analysis. That said, one of the key strengths of scoping reviews is to chart an outline of all existing research and advancements for a given field. Hopefully the changes to lines 30-42 on pg.4 clarify that the comparison of empirical research against financial burden across countries is a secondary objective of the review. This has the potential to guide future empirical work, and there is a precedent for such an exercise in the field of cost-utility analyses: https://journals.sagepub.com/doi/abs/10.1177/0272989X05276853
Methods and analysis Methods are explained in detail. The authors may clarify the following points:  1) Why was a scoping review chosen (not a systematic)? 2) Your choice of framework (Arksey and O'Malley) seems fitting. However, please clarify why this was chosen. Also, why have you chosen to not use the PRISMA guidelines (https://annals.org/aim/fullarticle/2700389/prisma-extensionscoping-reviews-prisma-scr-checklist-explanation). I think you address most of their points, but it would interest the reader to understand the rationale for your chosen approach. 3) When you use the Arksey and O'Malley framework, please clarify how it is special (and why it is relevant for your study) 4) Stage 2: Adding information from the grey literature seems fitting. In the paper you list highly relevant institutions. To explicitly allow others to understand which grey literature you search, I would suggest that you focus of key publications from these institutions (on UHC and/or FRP). 5) Study timeline: The section needs to be revised before publication of the protocol. 6) Is it one person who will conduct the analysis and data extraction? If so, please clarify. 	 1) As detailed in the paper, the broad objectives of the review lend themselves well to a scoping study which provides an outline of existing research on a given topic. Systematic reviews tend to have narrow objectives focussed on a small part of a research field, whereas all the research on the measurement of FRP must be reviewed in order to assess which methodologies are most transferable to allocative efficiency analyses. As a result, systematic reviews often provide an answer to a narrow research question, whereas scoping reviews tend to highlight what has been done/is known in a given field of study and the gaps in literature for future research. In addition to the main reason above, a number of other criteria of the proposed objectives and review match closer to a scoping review rather than a systematic one (please see the useful 'Table 1' in https://doi.org/10.1093/pubmed/fdr015 which compares scoping/systematic reviews across 6 aspects). 2) Thank you very much for sharing the newly developed PRISMA guidelines for scoping reviews. The checklist was not mentioned in the initial draft because the manuscript was submitted in September and the checklist was only published the following month in October. We have added a sentence (pg.5, lines 8-9) to clarify that the review will indeed seek to meet each of the 22 criteria contained in the checklist. 3) Arskey and O'Malley's framework, and subsequent literature that has contributed with additional recommendations (Levac et al. mentioned in the paper), are the recommended approach for conducting scoping reviews until alternative frameworks are developed. Please see for example Colquhoun et al. 2014: https://doi.org/10.1016/j.jclinepi.2014.03.013.

	4) We hope the rewording of the sentence referred to (pg.6, lines 2-3) clarifies that only literature on UHC and FRP will be scanned from the institutions named among others. 5) The timeline has been revised accordingly (please see pg.8, lines 14-19). 6) Given that this review is part of the corresponding author's PhD project, he will be solely responsible for the data extraction and analysis. That said, every stage of the process has and will be checked by his supervisors who are named co-authors. Please see lines 42-45 added in pg.8.
Discussion Well-written. You may also provide the reader with a discussion of the strengths and limitations of the study.	We agree and have added in a discussion of the key strengths and limitations of the proposed review (pg. 8 lines 38-46; pg. 9 lines 1-2).